

# Forecasting influenza epidemics by integrating internet search queries and traditional surveillance data with the support vector machine regression model in Liaoning, from 2011 to 2015

Feng Liang[1], Peng Guan[1], Wei Wu[1] and Desheng Huang[1,2]

[1] Department of Epidemiology, School of Public Health, China Medical University, Shenyang, Liaoning, China
[2] Department of Mathematics, School of Fundamental Sciences, China Medical University, Shenyang, Liaoning, China

## ABSTRACT

**Background:** Influenza epidemics pose significant social and economic challenges in China. Internet search query data have been identified as a valuable source for the detection of emerging influenza epidemics. However, the selection of the search queries and the adoption of prediction methods are crucial challenges when it comes to improving predictions. The purpose of this study was to explore the application of the Support Vector Machine (SVM) regression model in merging search engine query data and traditional influenza data.

**Methods:** The official monthly reported number of influenza cases in Liaoning province in China was acquired from the China National Scientific Data Center for Public Health from January 2011 to December 2015. Based on Baidu Index, a publicly available search engine database, search queries potentially related to influenza over the corresponding period were identified. An SVM regression model was built to be used for predictions, and the choice of three parameters ($C$, $\gamma$, $\varepsilon$) in the SVM regression model was determined by leave-one-out cross-validation (LOOCV) during the model construction process. The model's performance was evaluated by the evaluation metrics including Root Mean Square Error, Root Mean Square Percentage Error and Mean Absolute Percentage Error.

**Results:** In total, 17 search queries related to influenza were generated through the initial query selection approach and were adopted to construct the SVM regression model, including nine queries in the same month, three queries at a lag of one month, one query at a lag of two months and four queries at a lag of three months. The SVM model performed well when with the parameters ($C = 2$, $\gamma = 0.005$, $\varepsilon = 0.0001$), based on the ensemble data integrating the influenza surveillance data and Baidu search query data.

**Conclusions:** The results demonstrated the feasibility of using internet search engine query data as the complementary data source for influenza surveillance and the efficiency of SVM regression model in tracking the influenza epidemics in Liaoning.

Corresponding author
Desheng Huang,
dshuang@cmu.edu.cn

## INTRODUCTION

Seasonal influenza is a serious public health problem and remains rampant across the world. According to the latest estimates from the United States Centers for Disease Control and Prevention (US-CDC), there are about three to five million cases of severe illnesses, and about 2.9 to 6.5 million deaths each year caused by influenza epidemics (*World Health Organization, 2017, 2018*). National Health and Family Planning Commission of the People's Republic of China reported that China has 456,718 influenza cases with the incidence rate of 33.0994 per 100,000 in 2017 (*National Health and Family Planning Commission of the People's Republic of China, 2018*). Influenza epidemics pose significant social and economic challenges in China (*Yang et al., 2015*; *Wang et al., 2015*). It is necessary to establish a real-time flu surveillance system for rapid and effective responses in China.

A national noticeable infectious disease reporting system has been established to continuously report the influenza cases in China, while the system reports the flu activity one month before, putting the flu data lagged for a month. Traditional flu surveillance methods for prediction were mainly based on hospital or laboratory data (*Wang et al., 2017*). The idea of applying internet search query data for the infectious diseases prediction was from *Ginsberg et al. (2009)*, who presented a brand-new method providing nearly real-time surveillance of influenza-like illness and overcoming the limitations of lag-time in the traditional flu surveillance systems of the United States. Online search query data have a stronger tendency and immediacy and can maintain full synchronization with the flu epidemic. In addition, internet search query data can be measured in real time. In order to monitor the infectious diseases activity in time, numerous studies have been emerging recently based on online search query data or social media data, including Google (*Seo & Shin, 2017*; *Yang et al., 2017*; *Xu et al., 2017*; *Pollett et al., 2017*), Yahoo (*Polgreen et al., 2008*), Naver (*Shin et al., 2016*), Daum (*Woo et al., 2016*; *Seo et al., 2014*), Baidu search engine (*Guo et al., 2017b*), Twitter (*Wagner et al., 2017*; *Kagashe, Yan & Suheryani, 2017*; *Allen et al., 2016*; *Yun et al., 2016*) and Weibo (*Fung et al., 2013*; *Zhang et al., 2015*) social media, Wikipedia (*Hickmann et al., 2015*; *McIver & Brownstein, 2014*), hospital or clinicians' database (*Bouzille et al., 2018*; *Santillana et al., 2014*), and so on. As Google has been pulled out of mainland China in 2010, Google search query data and Google Flu Trends cannot be accessible in mainland China. This article will construct a forecasting model for influenza based on the ensemble data integrating traditional influenza cases data and Baidu search data, which is the most popular search engine in China.

Support vector machines (SVMs) are supervised learning models with associated learning algorithms, the application of SVM to classification and regression has been a hot topic recently. For solving the regression problem, SVMs have been applied to many fields: air quality forecasting (*Liu et al., 2017a*), water demanding and water quality prediction (*Ghalehkhondabi et al., 2017*; *Zhang, Zou & Shan, 2017*), biomedicine (*Nickerson et al., 2016*), etc. SVMs can efficiently perform a non-linear classification which is based on the

kernel trick, the inputs can be implicitly mapped into high-dimensional feature spaces. *Lampos et al. (2015)* indicated that a nonlinear query modeling approach presented the lowest cumulative nowcasting error. *Woo et al. (2016)* found that the SVM regression model based on weekly influenza incidence data and query data from the Korean website Daum performed well. *Guo et al. (2017b)* comprehensively assessed six machine learning algorithms based on Baidu search engine data and Dengue case data in Guangdong proposed that SVM regression model had a better performance than other forecasting techniques. Thus, this article attempted to build a SVM regression model to predict the flu activity.

The influenza epidemic situation varies greatly among different regions. China has a vast territory that spans tropical, subtropical and temperate regions, and it is a large challenge to establish an influenza prediction mechanism in the whole country. Liaoning province is located in both a coastal and bordered region in the northeastern part of China, where the feasibility of influenza prediction models based on internet search query data is still unknown. Thus, the purpose of this study was to investigate whether an early warning model utilizing both online influenza query data and traditional surveillance data could improve influenza prediction.

## MATERIALS AND METHODS

### Study setting and data collection

Liaoning is a coastal province in the northeast of China with a population of approximately 43.77 million in 2016 and a temperate continental monsoon climate. Official monthly reported number of influenza cases in Liaoning province in China was acquired from China National Scientific Data Center for Public Health (http://www.phsciencedata.cn) from January 2011 to December 2015. China National Scientific Data Center for Public Health is open for those registered users in mainland China and the latest influenza incidence data was the data in December 2015. Based on Baidu Index (http://index.baidu.com), an online keyword research tool which is publicly open for the public across the globe, search queries potentially related to influenza over the corresponding period were identified. "Influenza" was first adopted as a primary indicator term to find more related queries about influenza on the Chinese website (http://tool.chinaz.com/baidu/words.aspx). The website is a free online platform that providing internet keyword mining of the Baidu search engine in mainland China. Monthly average volume of those related search queries from Liaoning was extracted from Baidu Index website, from January 2011 to December 2015.

### Statistical analysis

The related influenza search terms were ranked first and those terms that having no data within one calendar year during the study period were excluded. Pearson correlation analysis was performed to explore the correlation between influenza-related search queries and the reported number of influenza cases in Liaoning. Those search terms that with the statistically significant correlation coefficient above 0.4 were sent to the construction of SVM regression model. The selection of maximum cross-correlation coefficient has been proposed in previous studies (*Guo et al., 2017a*; *Yuan et al., 2013*).

The influenza case surveillance data was divided into two parts, the fitting dataset and the validation dataset in SVM regression model. Forty-five months' data from January 2011 to September 2014 was used for model training, and the rest 15 months' data from October 2014 to December 2015 was used as the test set for model prediction. The choice of three parameters ($C$, $\gamma$, $\varepsilon$) in the SVM regression model was determined by leave-one-out cross-validation (LOOCV) during the model construction process. Three metrics were adopted to measure the performance of the SVM regression model, including Root Mean Square Error (RMSE), Root Mean Square Percentage Error (RMSPE), and Mean Absolute Percentage Error (MAPE). These three metrics are measures of prediction accuracy of a forecasting method in statistics. RMSE is very sensitive to the extreme errors or very small errors in a set of measurements, therefore RMSE can well reflect the precision of the forecasting. RMSPE is a percent difference between predicted and true values. MAPE is the most common measure of forecast error and it functions best when there are no extremes to the data (including zeros). The definitions of these three metrics are provided below. The notation in the study is as follows: $y_i$ denotes the observed value of the influenza cases at time $t_i$, $\hat{y}_i$ denotes the predicted value by SVM regression model at time $t_i$.

Root Mean Squared Error, a measure of the difference between predicted and true values, is defined as:

$$\text{RMSE} = \sqrt{\frac{1}{n}\sum_{i=1}^{n}(y_i - \hat{y}_i)^2} \tag{1}$$

Root Mean Square Percentage Error, a measure of the percent difference between predicted and true values, is defined as:

$$\text{RMSPE} = \sqrt{\frac{1}{n}\sum_{i=1}^{n}\left(\frac{y_i - \hat{y}_i}{y_i}\right)^2} \times 100\% \tag{2}$$

Mean Absolute Percentage Error, is the mean or average of the absolute percentage errors of forecasts and is defined as:

$$\text{MAPE} = \frac{1}{n}\sum_{i=1}^{n}\left|\frac{y_i - \hat{y}_i}{y_i}\right| \times 100\% \tag{3}$$

The statistical analysis and the construction of SVM regression model were performed using R statistical software version 3.4.2 with package e1071.

# RESULTS

## Baidu search terms filtering

According to the filtering criteria, there were 46 search terms left due to the available sequential data within one calendar year during the study period (Table S1). The Pearson correlation analysis was made between search terms and influenza cases across different lag periods (in the same month, at a lag of one month, at a lag of two months and three months). The correlation value of search terms with the influenza cases in the non-flu

**Table 1 Pearson correlation coefficients between search terms from Baidu search engine and the number of influenza cases in Liaoning, for the time period January 2011–December 2015.**

| Search terms | The same month | Lag one month | Lag two months | Lag three months | Non-flu season |
|---|---|---|---|---|---|
| Flu | 0.608** | 0.536** | 0.500** | 0.395** | 0.672** |
| Flu symptoms | 0.618** | 0.374** | 0.273* | 0.04 | 0.430* |
| Influenza type A | 0.489** | 0.134 | 0.005 | −0.287* | 0.048 |
| Influenza vaccine | 0.259* | 0.436** | 0.645** | 0.764** | 0.328 |
| Is it necessary to get vaccinated against the flu? | 0.103 | 0.362** | 0.626** | 0.814** | 0.255 |
| Flu virus | 0.621** | 0.435** | 0.282* | 0.175 | 0.711** |
| The symptom of flu | 0.656** | 0.438** | 0.209 | −0.054 | 0.235 |
| Influenza drugs | 0.639** | 0.431** | 0.218 | −0.116 | 0.337 |
| The symptoms of type A flu | 0.157 | 0.029 | −0.061 | −0.225 | 0.320 |
| Prevent flu | 0.623** | 0.644** | 0.663** | 0.511** | 0.374* |
| Swine flu | 0.371** | 0.290* | 0.172 | 0.075 | 0.459* |
| H1N1 flu | 0.021 | −0.157 | −0.231 | −0.409** | −0.302 |
| Beijing flu | 0.249 | 0.055 | 0.037 | −0.148 | 0.313 |
| Swine flu symptoms | 0.032 | 0.142 | −0.104 | −0.101 | −0.039 |
| How to prevent flu | 0.023 | −0.025 | 0.056 | 0.007 | −0.006 |
| Viral flu | 0.484** | 0.339** | 0.234 | −0.023 | 0.587** |
| How to prevent flu | 0.129 | 0.087 | 0.201 | 0.125 | −0.014 |
| Spanish flu | 0.459** | 0.479** | 0.405** | 0.409** | 0.491** |
| Flu prevention | 0.178 | 0.012 | −0.092 | −0.133 | 0.380* |
| Side effects of flu vaccine | 0.084 | 0.379** | 0.657** | 0.804** | 0.254 |
| The prevention measures of flu | −0.079 | −0.056 | 0.043 | 0.061 | −0.108 |
| Type A H1N1 flu | 0.089 | −0.024 | −0.281* | −0.362** | −0.379* |
| Flu therapy | 0.438** | 0.183 | 0.168 | −0.075 | 0.291 |
| The prevention of flu | −0.335** | −0.357** | −0.251 | −0.285* | −0.394* |
| Influenza epidemic | 0.09 | −0.075 | −0.207 | −0.375** | 0.108 |
| Influenza vaccine price | −0.273* | −0.185 | −0.002 | 0.126 | −0.124 |
| Type A flu | 0.383** | 0.326* | 0.025 | −0.193 | 0.508** |
| Type A flu virus | 0.586** | 0.395** | 0.276* | 0.002 | 0.369* |
| New type of flu | 0.352** | 0.430** | 0.037 | −0.076 | 0.387* |
| Type A influenza | 0.016 | −0.222 | −0.138 | −0.345** | −0.197 |
| Love flu strain | 0.054 | −0.165 | −0.166 | −0.296* | −0.425* |
| Flu concept stock | 0.266* | 0.226 | 0.011 | −0.072 | 0.349 |
| Seasonal influenza | −0.046 | −0.122 | −0.12 | −0.172 | 0.515** |
| Love flu | −0.300* | −0.310* | −0.350** | −0.368** | −0.114 |
| Type A H1N1 flu virus | −0.109 | −0.22 | −0.24 | −0.131 | −0.160 |
| New flu | −0.048 | −0.064 | −0.151 | −0.25 | −0.016 |
| How to treat swine flu | 0.223 | 0.547** | 0.476** | 0.381** | 0.200 |
| Influenza transmission route | 0.228 | 0.1 | −0.09 | −0.076 | 0.346 |
| The route of transmission of flu | 0.216 | −0.011 | −0.12 | −0.164 | 0.207 |
| Treatment program of A type H1N1 flu | −0.145 | −0.19 | −0.24 | −0.304* | 0.102 |

(Continued)

| Search terms | The same month | Lag one month | Lag two months | Lag three months | Non-flu season |
|---|---|---|---|---|---|
| Flu (space) symptom | 0.147 | 0.042 | −0.068 | −0.157 | 0.358 |
| H1N1 flu symptom | 0.346** | 0.196 | 0.141 | 0.064 | 0.260 |
| Sheep flu | 0.011 | −0.038 | −0.053 | −0.107 | 0.211 |
| Super flu | 0.314* | 0.248 | 0.133 | 0.16 | 0.172 |
| The symptom of swine flu | 0.134 | 0.071 | −0.025 | −0.124 | 0.341 |
| Taiwan flu | 0.124 | 0.066 | 0.104 | −0.188 | 0.374* |

Notes:
* Indicates the *P* value with statistically significance at 0.05 level.
** indicates the *P* value with statistically significance at 0.01 level.

**Table 2 Strongly correlated search terms with the number of influenza cases in different lag periods.**

| Lag time | Search keywords |
|---|---|
| The same month | Flu, flu symptoms, influenza type A, flu virus, the symptoms of flu, influenza drugs, viral flu, flu therapy, type A flu virus |
| Lag one month | Spanish flu, new type of flu, how to treat swine flu |
| Lag two months | Prevent flu |
| Lag three months | Influenza vaccine, is it necessary to get vaccinated against the flu, H1N1 flu, side effects of flu vaccine |

season is provided as a basal level of their relationship (Table 1). Twenty-nine of the remaining 46 terms were excluded, because their Pearson correlation coefficients between the search terms and influenza cases were less than 0.4 across every lag period. A total of 17 search terms which were strongly correlated with influenza cases across different lag periods were retained for the construction of SVM regression model, including nine queries in the same month, three queries at a lag of one month, one query at a lag of two months and four queries at a lag of three months (Table 2). Meanwhile, the amount of influenza cases might have an impact on the amount of incident cases of the following months. The Pearson correlation analysis was performed to compare the relationship between the reported number of influenza cases of the month and historically reported number of influenza cases. The correlation coefficients were 0.672, 0.498 and 0.151 at the lag time of one month, two months and three months, respectively. The reported number of influenza cases at the lag time of one month has shown the strongest correlation, thus it was submitted to SVM regression model.

## Parameter selection of SVM regression model

The mathematical formula of SVM regression model is provided below: $\alpha$, $\alpha^*$ are Lagrangian operator. $C$ is the upper bound of all variables, $Q$ is a $k$ by $k$ positive semidefinite matrix, $Q_{ij} = y_i y_j K(x_i, x_j)$, and $K(x_i, x_j)$ is the kernel.

$$\min_{\alpha, \alpha^*} \frac{1}{2}(\alpha - \alpha^*)^{\mathrm{T}} Q(\alpha - \alpha^*) + \epsilon \sum_{i=1}^{k}(\alpha_i - \alpha_i^*) + \sum_{i=1}^{k} y_i(\alpha_i - \alpha_i^*)$$
$$\text{s.t.} \, 0 \leq \alpha_i, \alpha_i^* \leq C, i = 1, \dots, k, \tag{4}$$
$$\sum_{i=1}^{k}(\alpha_i - \alpha_i^*) = 0.$$

**Table 3 The SVM model precision of different $C$ values ($\gamma = 0.05556$, $\varepsilon = 0.1$).**

| $C$ | Training error | Test error |
|---|---|---|
| 0.0001 | 8,812.675 | 8,834.564 |
| 0.001 | 8,768.452 | 8,806.329 |
| 0.01 | 8,363.176 | 8,532.467 |
| 0.1 | 5,831.012 | 6,661.826 |
| 1 | 1,4647.06 | 4,052.645 |
| 2 | 498.3551 | 3,900.983 |
| 3 | 215.0484 | 4,003.317 |
| 4 | 175.4402 | 4,116.998 |
| 5 | 147.7603 | 4,215.681 |
| 10 | 76.7374 | 4,756.99 |
| 100 | 71.55703 | 4,792.467 |

**Table 4 The SVM model precision of different $\gamma$ values ($C = 1$, $\varepsilon = 0.1$).**

| $\gamma$ | Training error | Test error |
|---|---|---|
| 0.0001 | 8,130.444 | 8,199.893 |
| 0.001 | 4,533.864 | 4,933.536 |
| 0.005 | 1,932.273 | 3,239.05 |
| 0.01 | 1,604.143 | 3,502.44 |
| 0.02 | 1,493.212 | 3,655.345 |
| 0.03 | 1,465.351 | 3,773.955 |
| 0.04 | 1,466.39 | 3,873.576 |
| 0.05 | 1,459.852 | 3,982.198 |
| 0.1 | 1,470.783 | 4,796.493 |
| 1 | 2,441.952 | 7,936.262 |

The expression of radial basis function is provided below:

$$K\left(x_i, x_j\right) = \exp\left\{-\gamma\left|x_i - x_j\right|^2\right\} \tag{5}$$

During the process of leave-one-out cross-validation, we started from the default value ($\gamma = 0.0556$, $\varepsilon = 0.1$), then we adjusted the $C$ value to observe the model fitting results (Table 3). The same method was applied to the selection of the other two parameters, $\gamma$ and $\varepsilon$ (Tables 4 and 5). The values of these three parameters were evaluated according to the lowest test error, then the optimal parameters of the model were determined ($C = 2$, $\gamma = 0.005$, $\varepsilon = 0.0001$).

## Comparison and prediction of SVM regression models from different data sources

Compared with the model based on influenza case data at the lag time of one month and the source of Baidu search data, the SVM regression model based on ensemble data

Liang et al. (2018), *PeerJ*, DOI 10.7717/peerj.5134

**Table 5 The SVM model precision of different ε values ($C = 1$, $\gamma = 0.05556$).**

| ε | Training error | Test error |
|---|---|---|
| 0.0001 | 1,388.189 | 3,985.145 |
| 0.001 | 1,388.479 | 3,986.231 |
| 0.01 | 1,392.287 | 3,993.167 |
| 0.05 | 1,412.391 | 4,027.2 |
| 0.08 | 1,440.697 | 4,051.417 |
| 0.09 | 1,452.264 | 4,052.576 |
| 0.1 | 1,464.717 | 4,052.645 |
| 0.2 | 1,539.904 | 4,060.86 |
| 0.3 | 1,709.969 | 4,153.16 |
| 0.4 | 1,965.82 | 4,346.04 |
| 0.5 | 2,361.972 | 4,643.752 |
| 1 | 5,522.041 | 7,408.229 |

**Table 6 Similarity metrics between 3 data sources: the number of influenza cases at a lag of one month, Baidu keywords, ensemble data, for the time period October 2014–December 2015.**

| | RMSE | RMSPE (%) | MAPE (%) |
|---|---|---|---|
| Influenza cases at a lag of one month | 82.874 | 40.658 | 35.150 |
| Baidu keywords | 43.472 | 30.438 | 26.806 |
| Ensemble model | 42.654 | 29.687 | 26.197 |

integrating historical influenza surveillance data and Baidu search data showed the best accuracy with lowest RMSE (42.654) and best robustness with lowest MAPE (26.197%), as seen in Table 6.

The predicted values of the above three models and the actual number of influenza cases from October 2014 to December 2015 have been presented in Fig. 1. It was easily to find that the SVM model prediction's curve was almost identical when comparing the model based on internet search query data with the model based on ensemble data, and trend of the curve were consistent with the overall development trend of the actual influenza cases curve. The SVM regression model based on ensemble data was capable of predicting the timing and magnitude of most periods, whereas it failed to predict the influenza outbreak peak in March 2015. The predictions of the model based on flu data at the lag time of one month were significantly lower than the actual value in the previous six-month forecasting, but the overall trend was consistent in the following nine months. The residual of each predictor is displayed in Fig. 2.

## DISCUSSION

This article presented an efficient SVM regression model to predict flu activity and track the epidemic orbit in Liaoning province of China. The entire analysis demonstrated that the SVM regression model based on ensemble data was better than the model based on

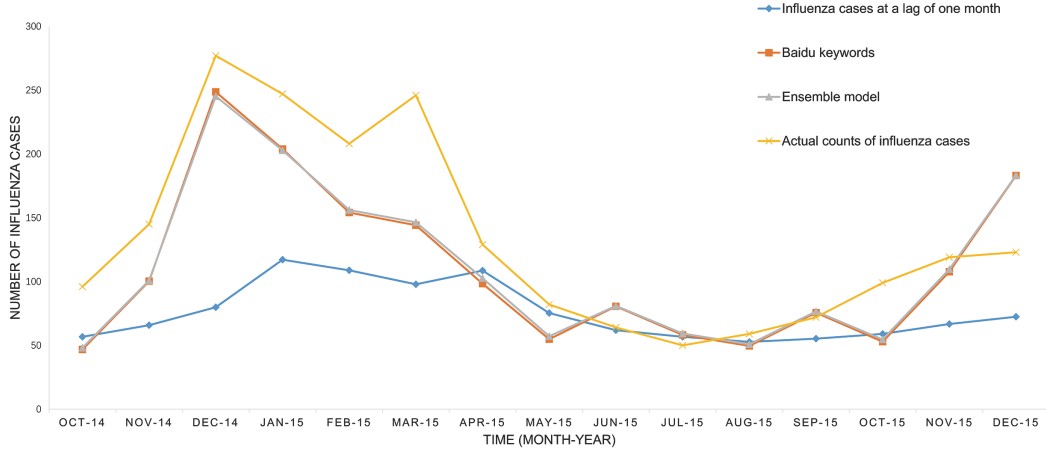

**Figure 1** The performance of the three available predictors.

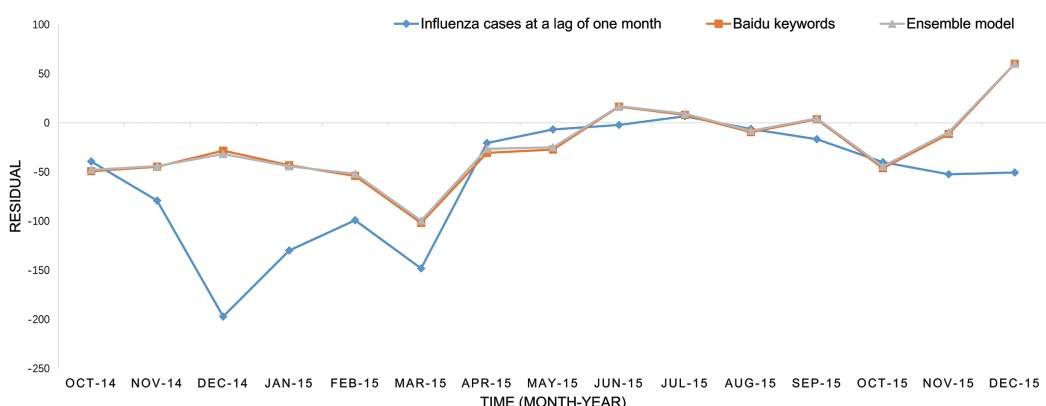

**Figure 2** The residuals of the three predictors.

conventional data. It could be a complement to the traditional surveillance for influenza dynamics.

With the rapid development and popularity of the Internet, this new method of infectious diseases surveillance system based on online search query data is more convenient and accurate. According to the 41st Statistical Report on Internet Development released by China Internet Network Information Center (CINIC), the internet users in China have steadily increased and up to 772 million, and the internet popularity rate reached 55.8%, exceeding the global average level until December 2017 (*China Internet Network Information Center, 2018*). Baidu search engine is the most widely accepted search engine in China, making it the most representative and available data source for the studies targeting tracking the online seeking behavior of Chinese people. Based on Baidu search query data, Chinese scholars have made great efforts in the field of disease monitoring, such as Norovirus (*Liu et al., 2017b*), Dengue (*Li et al., 2017*), Hand, foot, and mouth disease (*Du et al., 2017*), and epidemic erythromelalgia (EM) (*Gu et al., 2015*). These forecasting models got great performances in the field of early warning.

However, most of the researches focused on southeastern coastal regions of China, such as Guangdong and Zhejiang, and few disease prediction models was constructed and applied in the coastal areas in the northeast China. It is a significant attempt to predict the influenza activity in Liaoning province located in the northeast of China. This article could provide some hints and lessons for the flu forecasting and alerting in the Northeast of China.

Strong correlation between influenza cases and search terms of Baidu was found in the present study. Influenza is characterized by a short incubation period and a sudden onset of symptoms such as fever, cough (usually dry), etc., and it is reasonable that most of search terms about flu virus, symptoms and therapy closely correlated with influenza cases at the same month. The most effectively vaccine injection timing is about one to two months prior to the flu season. Winter and spring are the peak flu seasons in Liaoning, China, thus September and October are the best months for flu vaccination in the study area. The search behavior about flu vaccine is often earlier than the vaccination timing, so we found that the search terms of Baidu at a lag of three months had a strong correlation with the occurrence of influenza cases.

The present study showed that the forecasting model based on internet search query was better than the model based on traditional data in terms of accuracy and stability. The results were consistent with the results of other studies (*Guo et al., 2017b*; *Yuan et al., 2013*). However, *Olson et al. (2013)* investigated the reliability of Google Flu Trends (GFT) of 2003 to 2013 and compared the flu timing and intensity between forecasting data and actual influenza incidence at the national, regional and local levels. They concluded that GFT data could not serve as the reliable surveillance for seasonal or pandemic influenza and traditional surveillance are still irreplaceable. The main reason was that GFT was based on internet data without considering the epidemiological factors such as the age distribution of patients, geographical location, illness complaints or clinical manifestations. Our study proved the advantages of ensemble source data integrating traditional influenza incidence data and search engine data in the field of forecasting. Meanwhile, there may be some space to improve the SVM model presented in the present study. Although most of the forecasting values were fitted well with actual influenza cases in the SVM regression model, they failed to identify the influenza's peak in March 2015. The climatic factors have great impact on Influenza incidence (*Gomez-Barroso et al., 2017*), thus the possible reason of their missing might be that March are the cold month and the flu peak seasons in northeastern China while internet search query data could not distinguish the situation.

Several limitations in influenza forecasting model based on ensemble data integrating traditional influenza cases data and Baidu search engine data need to be mentioned. Firstly, media report may influence the internet searching behavior, which will have an impact on the performance of forecasting model directly. In addition, without considering the impact factors of influenza, such as seasonal and meteorological factors, the forecasting results may have bias to some degree. Furthermore, correlation analysis of the search keywords mainly was based on previous vocabularies data. However, in pace with the rapid changes of the internet environment, many fresh online search vocabularies produced at every moment. The fresh vocabularies were hard to be tracked and usually have been overlooked.

## CONCLUSIONS

The present study built a forecasting model based on ensemble data integrating Baidu search query data and traditional flu data in Liaoning province. The model based on ensemble data showed the best accuracy and best robustness in SVM regression model, rather than the models based on other single data sources. It could be a complement of the traditional surveillance for influenza dynamics in Liaoning.

## ACKNOWLEDGEMENTS

This study was partly based on the information from the data provided by China National Scientific Data Center for Public Health and Baidu index website.

### Funding

This work was supported by the National Natural Science Foundation of China (No. 71573275). There was no additional external funding received for this study. The funders had no role in study design, data collection and analysis, decision to publish, or preparation of the manuscript.

### Grant Disclosures

The following grant information was disclosed by the authors:
National Natural Science Foundation of China: 71573275.

### Competing Interests

The authors declare that they have no competing interests.

### Author Contributions

- Feng Liang performed the experiments, analyzed the data, prepared figures and/or tables, authored or reviewed drafts of the paper, approved the final draft.
- Peng Guan performed the experiments, prepared figures and/or tables, approved the final draft.
- Wei Wu performed the experiments, approved the final draft.
- Desheng Huang conceived and designed the experiments, performed the experiments, analyzed the data, contributed reagents/materials/analysis tools, authored or reviewed drafts of the paper, approved the final draft, secured funding.

### Data Availability

 The raw data and R codes are provided in the Supplemental Files.

### Supplemental Information

Supplemental information for this article can be found online at http://dx.doi.org/10.7717/peerj.5134#supplemental-information.

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
