# Peer review of "Forecasting influenza epidemics by integrating internet search queries and traditional surveillance data with the support vector machine regression model in Liaoning, from 2011 to 2015"

_PeerJ, doi:10.7717/peerj.5134_

## Round 0.1 · original submission · Minor Revisions

All three reviewers concur that more details have to be provided for methods, and a more precise description is also needed for some ideas. Please respond to all the comments of the reviewers.

·

Basic reporting

There are a clear manuscript , written in English, this manuscript cover the professional standart of courtesy and expression, the references are sufficient and correct.

Main contruction of the manuscript is aceptable

Main objectives is clear too

Experimental design

Dessign is appropiate, to solve the research question

Rigorous research methods were designed

Authors must be more clear about how internet data were obtained.

Validity of the findings

Data is robust, statistically sound and controlled

Additional comments

Please give more details about who and how internet information was obtained. Additionally time and permisions to use the platform

·

Basic reporting

Liang et al. Developed and explored the utility of a SVM regression model in the forecasting influenza epidemics by integrating data from internet search queries and data from traditional surveillance. The intention of this work is relevant, because it could improve the strategies for attending expected outbreaks or epidemics.
Authors have found relevant words that have relation with the number of influenza cases in four times: in the same month, Lag one month, lag two months and lag three months, however they don don’t explain why those times were the most relevant. In this sense it should be useful to include in the table 1 the correlation values of search terms in a time in where the number of influenza cases is the lowest. This could show how the search frequency is when using these words "at a basal level", that is, at a time not related to the high frequency of influenza cases, or during national preparations to face seasonal influenza (campaigns or preparations). of prevention).
Figures 1 and 2 does not contain enough legends to understand it, also it does not have titles in the axis and units. I recommend including that missing data to improve the figures and to avoid misinterpretation.

Experimental design

Some important data could improve the work
1.- [Lines 121-138]
There is not clear, why the RMSE, RMSPE and MEPE metrics are adequate to measure the performances of the SVM regression model.
A more detailed explain for the use of this metrics could strength the experimental design. it should be more important to show the epidemiological utility of this metrics instead of the mathematical fundamentation.
2.- [Lines 103-113]
“Official monthly reported number of influenza cases in Liaoning province, China was acquired from China National Scientific Data Center for Public Health (http://www.phsciencedata.cn) from 107 January 2011 to December 2015”
Why did you include those period of time? Why you did not include data from 2016 to 2017?
There are some weakness in the design:
1.- Lines [123-124] “the rest 15 months’ data from October 2014 to December 2015 was performed for model validation”
Why did you only include 15 months for validation? I mean if you nowadays have recent data, why did you only use 15 months for validation? You could strength your SVM regression model if you extend the period of validation from October 2014 to December 2017.

Validity of the findings

Although some tables and figures show relevant results, there are some months in where there is not a good correlation between the ensemble model and the influenza cases (For example, two peaks of cases did not were forecasted, but also there were months that the model forecasted an increased number of cases and the real number was lower). Those findings not necessary imply that the model fail to detect tendencies or even a possible epidemic. If you could extend at least three years the period of validation, there were more available data to know the utility of the proposed model.

Additional comments

no comment

·

Basic reporting

1. In abstracts, the reported parameters "C=4, ε=0.01, γ=0.04" are out of context. It is unclear what these parameters are referring to.
2. Standard acronym for United States Centers for Disease Control and Prevention is "US-CDC"
3. The model labeled "Past influenza cases" are not clearly stated in the article. My best guess is from line 178 that same SVM model is applied with data of past influenza cases, and it is unclear how many lags are used.

Experimental design

1. In line 119, the requirement for search terms to have correlation coefficient above 0.4 seems arbitrary. What's the justification for this "0.4"requirement?
2. In line 158, for clarity, the mathematical formula and the parametrization of SVM should be given.
3. In line 161 and Table 3, is cross-validation used to select parameter C? It seems authors are selecting C to minimize overall RMSE, which is prone to overfitting. The same comment applies to parameter γ and ε.

Validity of the findings

1. What's reason for study period January 2011 to December 2015? Is this due to data limitation?
2. Is there discussion on why search terms can have lagged impact as in Table 2? I found the vaccine terms intuitive and sensible, but found it not as convincing for lag one month search keywords.

Additional comments

The authors try to use Support Vector Machine for real-time nowcast of monthly reported number of influenza cases in Liaoning province, China. In general the article is well written and the results are convincing. However, there is some ambiguity in the methodology section that should be further clarified.

---

## Round 0.2 · accepted · Accept

Thank you for giving us the opportunity to review your work and congratulations once again for its acceptance.

·

Basic reporting

no comment

Experimental design

no comment

Validity of the findings

no comment

Additional comments

The authors try to use Support Vector Machine for real-time nowcast of monthly reported number of influenza cases in Liaoning province, China. Although the approach has been previously developed for infectious diseases surveillance in other countries, it is first time being applied to Liaoning province, China. The method is rigorous and findings are convincing, and in general the paper has good quality to join the scholarly literature. PeerJ seems a perfect outlet for this.